# An automated platform to monitor long-term behavior and healthspan in *Caenorhabditis elegans* under precise environmental control

Kim N. Le [1,4], Mei Zhan[1,4], Yongmin Cho [2,3,4], Jason Wan [1], Dhaval S. Patel [2] & Hang Lu [2✉]

Health and longevity in all organisms are strongly influenced by the environment. To fully understand how environmental factors interact with genetic and stochastic factors to modulate the aging process, it is crucial to precisely control environmental conditions for long-term studies. In the commonly used model organism *Caenorhabditis elegans*, existing assays for healthspan and lifespan have inherent limitations, making it difficult to perform large-scale longitudinal aging studies under precise environmental control. To address these constraints, we developed the Health and Lifespan Testing Hub (HeALTH), an automated, microfluidic-based system for robust longitudinal behavioral monitoring. Our system provides long-term (i.e. entire lifespan) spatiotemporal environmental control. We demonstrate healthspan and lifespan studies under a variety of genetic and environmental perturbations while observing how individuality plays a role in the aging process. This system is generalizable beyond aging research, particularly for short- or long-term behavioral assays, and could be adapted for other model systems.

[1] Wallace H. Coulter Department of Biomedical Engineering, Georgia Institute of Technology and Emory University, Atlanta, Georgia 30332, USA. [2] School of Chemical & Biomolecular Engineering, Georgia Institute of Technology, Atlanta, Georgia 30332, USA. [3]Present address: Department of Systems Biology, Harvard Medical School, Boston, MA 02115, USA. [4]These authors contributed equally: Kim N. Le, Mei Zhan, Yongmin Cho. ✉email: hang.lu@gatech.edu

Environmental factors play major roles in the health and longevity of an individual[1]. In nature, individuals are exposed to a wide variety and combination of environmental conditions. In the confines of a research laboratory, it is difficult to mimic this complexity and study a large range of environmental factors. To address these difficulties, researchers have leveraged model organisms to understand some of the effects of gene–environment interactions and stochasticity on the aging process[2–6]. For instance, examining the effect of select environmental cues on longevity in Caenorhabditis elegans has already uncovered important genetic pathways and regulators that provide insight into the effect of stressors on aging[7–10]. However, there are major challenges in providing precise environmental control for longitudinal studies on a large scale, even for C. elegans, making it difficult to study the long-term effects of environment on the aging process.

Thus, there is a need for a robust method to probe a wide variety of environmental conditions in a high-throughput manner. In addition, an experimental system needs to provide precise spatiotemporal control throughout the entirety of the organism's lifespan to reduce any unintended environmental fluctuations that would result in confounding biological noise. Existing methods for studying lifespan and the period of healthy living, or "healthspan", in C. elegans are limited and lack the ability to provide precise environmental control. Although traditional lifespan and healthspan assays have uncovered important insights into the aging process, they require large amounts of manual labor, making it difficult to scale for different genotypes or environmental conditions[11–14]. Recent high-throughput methods have been developed for scoring longevity or observing behavior over time[15–18]; however, they also lack precise control in regulating the organism's immediate environment. For instance, dietary restriction (DR), an evolutionarily conserved environmental perturbation that modulates aging, cannot be studied as food levels deplete over time within these closed-culture systems. Existing microfluidic devices for studying C. elegans lifespans allow for precise spatiotemporal environmental control, but often lack experimental robustness, longitudinal tracking, are limited to a range of low food levels, or have not feasibly demonstrated the ability to scale up for high-throughput studies[19–22].

Here we present the Health And Lifespan Testing Hub (HeALTH), an automated microfluidic-based platform for longitudinal behavioral monitoring during healthspan studies that allows for controlled dynamic changes during worm culture. Using custom-built hardware and software we can control the temperature and fluid conditions for over 1400 isolated individuals, allowing us to culture large populations under precise spatiotemporal environments in a high-throughput and scalable manner. We obtained longitudinal behavioral data for each individual from the L4-larval stage until its death, with sub-hourly resolution, giving us insight into functional decline over time. To demonstrate the versatility and robustness of the platform we obtained longevity and healthspan data for established long-lived (daf-2) and short-lived (daf-16) mutants[23]. We also examined the influence of several environmental factors, looking at the impact of DR and culture temperature, exploring how both steady-state and controlled temperature variations can impact aging in poikilotherms.

## Results

### Longitudinal monitoring under precise environmental control.

Traditional healthspan assays require periodic sampling and scoring of the worm's behavior to measure its functional decline during aging. Populations are kept within large incubators and cultured either in liquid bacterial suspensions or on plates with a seeded lawn of E. coli as a food source. However, the concentration of the bacterial food source depletes over time as the worms consume it, introducing uncontrollable environmental variability. Furthermore, population-level assays do not allow longitudinal tracking of individuals, confounding the role of stochastic factors in the aging process. To improve upon existing healthspan assays, we engineered a platform (Fig. 1a) that provides (1) automated longitudinal video recordings of behavior over time, (2) temperature control, and (3) a microfluidic culture environment for controlled exposure to food throughout time (Fig. 1b–c).

A major feature of our platform is the ability to acquire detailed behavioral data for many individuals throughout their entire lifespan. To accomplish this in both a cost and space-effective manner, our platform uses a time-shared camera placed on an automated x–y stage. The camera obtains 10 s videos of individuals twice an hour; the acquisition duration and frequency can easily be modified to accommodate multiple experiments being concurrently monitored on the system. The platform currently allows for up to 24 different experimental populations to be simultaneously cultured. This could easily be increased by multiplexing or through the use of smaller culture devices, providing experimental modularity and scalability. As a result, we can automate and readily modify our video acquisition, allowing us to easily scale-up experiments to perform large quantities of behavioral recordings.

To allow precise temporal control of temperature, we developed a Peltier-based, thermal control module. Temperature is a well-known factor for longevity in poikilotherms[9], so it is essential to maintain constant control of the temperature experienced by the worms over time. Typical assays require the repeated transfer of worms from temperature-controlled incubators to benchtop microscopes for phenotypic scoring, introducing frequent unintended thermal perturbations. C. elegans have been shown to detect small changes in temperature (0.05–0.1 °C), altering behavior in response to thermal fluctuations[24,25]. To avoid these undesired perturbations and to ensure that temperature is highly controlled over time and robust to environmental disturbances, we engineered a stable thermal module with a large thermal mass unit with feedback control. Temperature is sensed by an embedded thermocouple and a real-time PID controller ensures the output temperature to be within ±0.1 °C precision, maintaining the desired temperature throughout the lifespan of the worms on the order of weeks (Fig. 1d). Because of the real-time sensing and the controller modulating the Peltier element's work, we can also alter temperature at will and with precision, allowing us to examine how controlled dynamic perturbations, mimicking the natural environment, could impact aging. Due to the small size and modularity of these controllers, our system can maintain multiple temperature conditions on the same platform for experimental flexibility.

To longitudinally track individual behavior through the lifespan and to precisely control the food conditions each animal experiences, we developed a microfluidic device[26] coupled with a continuous food delivery system. A single device cultures up to 60 worms, each individually housed within a 1.5 mm diameter circular chamber (Fig. 1b–c, Supplementary Movies 1–15). Worms are loaded into the device at the L4 larvae stage, allowing us to longitudinally track individuals and observe how complex behaviors change throughout their entire adulthood. Each of the chambers is connected by a large channel that allows for fluid exchange and food delivery. Thus, we can control the local environment experienced by the worm, such as food abundance or drug compounds, and observe its effects on healthspan and longevity. To drive fluid exchange, we developed a custom-built

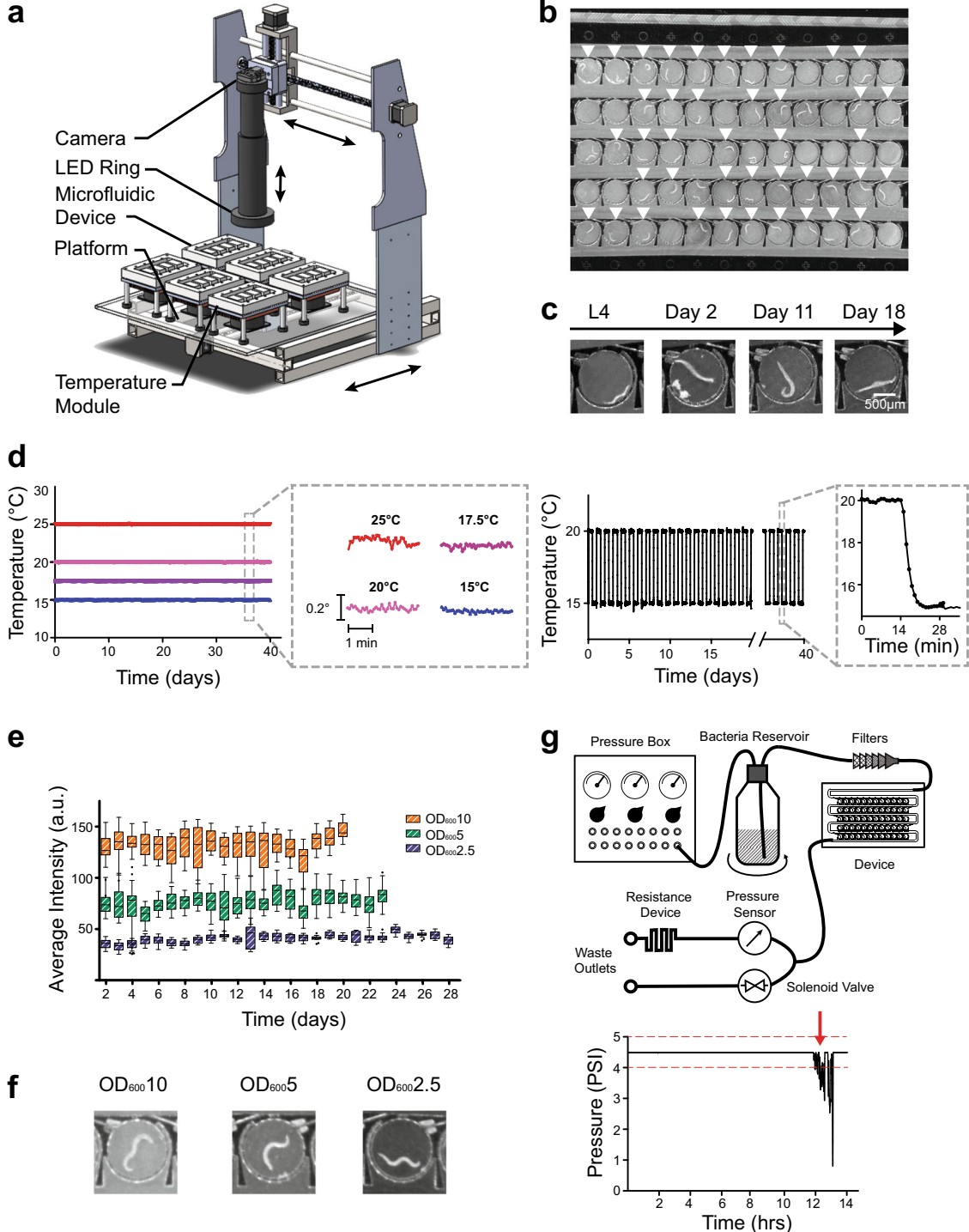

**Fig. 1 HeALTH enables robust, long-term culture of *C. elegans* under precise spatiotemporal environmental control. a** Overview of the platform. **b** Image of the microfluidic device with worms loaded in the circular chambers. Single-loaded chambers are indicated by white arrows. **c** Images of an individual monitored throughout their entire adult lifespan, from L4 stage to death at Day 18. **d** Common culture temperatures (15–25 °C) can be set and maintained for over a month with ±0.1 °C precision. Temperatures can also be dynamically controlled over time (ex. temperature swings from 15–20 °C every 12 h). **e** Tukey's boxplot of different food levels (OD$_{600}$10, 5, and 2.5), measured by average intensity within the device. **f** Images of worms under different food levels (OD$_{600}$10, 5, and 2.5). **g** Overview of the flow path and downstream pressure sensor for continuous flow of bacteria to the device and a representative example of a significant pressure drop (red arrow) indicating a clog upstream of the device.

pressure delivery system to provide continuous flow within the device, providing constant exposure to the desired food concentration (Fig. 1e–f). Furthermore, pressurized food delivery enables the easy scalability of food sources across multiple devices.

An additional benefit of continuous pressure driven flow is the ability to detect bacterial accumulation over time. A major challenge associated with long-term culture within a microfluidic device is the accumulation of bacterial biofilms over time, particularly at higher concentrations of bacteria. To address this

issue and improve experimental robustness, the system has a continuous pressure monitoring system downstream of the device. The sensor detects drops in pressure and flow rate experienced within the device, which indicates the presence of bacterial accumulation or clogs in the upstream filters (Fig. 1g). The system then alerts the experimenter to intervene. As a result, our platform can reliably culture worms under high bacterial concentrations for over 2 months, which has not been previously demonstrated in a microfluidic device[19–22]. This then allows us to robustly perform long-term experiments without introducing artifacts.

All hardware is integrated and controlled via a custom GUI, which enables experimental set-up, entering metadata, and monitoring the status of the worms as they age (Supplementary Fig. 1). The automated system reports real-time temperature and pressure data and generates approximately 15MB/s of video data for each device under the current acquisition conditions. Recorded videos are processed off-line to extract behavioral data. An overview of the design of each subsystem, along with a complete list of parts and cost to create the HeALTH system can be found in Supplementary Note 1.

**Behavioral analysis for insights into functional decline**. Because the system is capable of capturing terabytes of raw video data over the course of the entire lifespan of the monitored individuals, there is a need for a robust analysis pipeline that will be able to extract behavioral features with minimal user input. To handle the complex environmental conditions, we developed a custom MATLAB script that scores the lifespan of individuals and extracts key behavioral metrics to track longitudinal behavioral decline during the aging process (Fig. 2a). Briefly, we use computer vision to identify the location of each worm within the device for each video. We segment the body using a consensus approach informed by positional and morphological parameters, which are updated over time. We then measure behavioral features—such as the pixel difference across frames in a video (raw movement), centroid speed, amplitude of body bends, and the swimming frequency—and track them as the worms age. Worms are scored as dead if no raw movement is detected after a period of two days (Fig. 2b). We validated a sample of the lifespan results automatically generated with our code with a blinded manual curation of the videos and have found no significant difference between the two conditions (Supplementary Fig. 2).

**Recapitulating lifespan and behavior trends in mutants**. A major challenge when developing long-term culture systems is to preserve known longevity trends across populations and to ensure no adverse effects prematurely shorten the lifespan of the population. To validate our system, we measured the lifespan of a wild-type (N2) population, along with two different insulin/IGF-1 pathway mutants, *daf-2(e1368)* and *daf-16 (mu86)*, which are well-known long- and short-lived mutants respectively. Longevity assays performed in HeALTH are comparable to lifespan trends found using traditional plate assays, for both wild-type and mutant populations (Fig. 3a–b). Importantly, our platform exhibits similar variability (including trial-to-trial, across devices, and individual variation) across all replicates compared to plate-based trials, indicating similar experimental reproducibility (Supplementary Fig. 3, Supplementary Table 1)[27]. This ability to recapitulate both the trends and variability in comparison to traditional plate assays underscores the viability of using HeALTH to perform lifespan studies.

Next, we examined how health is impacted during aging by monitoring behavioral decline. Previous literature demonstrated the need to look at evoked behavior as an accurate measure of

behavioral decline during aging[13,16]. Our microfluidic device provides mechanical stimulus through fluid flow, allowing us to observe how stimulated locomotion changes as individuals age (Supplementary Fig. 4). Using our automated analysis pipeline, we can extract several behavioral features and track how they change over time, which can be observed for individuals and as a population average (Fig. 3c–d). There are distinct variations in behavioral decline across phenotypes and genotypes. For example, swimming frequency is sporadic due to the variety of behavioral phenotypes aside from swimming, particularly as the worm ages. By examining a variety of behavioral phenotypes, we can obtain a more holistic sense of decline, tracking how more defined movements (such as swimming) decline in comparison to rougher measures (such as raw movement).

Similar to previous studies, across all genotypes there appears to be a consistent initial period of frequent "high activity" movement, followed by a period of sporadic "low activity" movement until the individual's death[12–14,16]. Bouts of consistent movement suggest an initial period of good health for the worm, which may indicate a lack of sarcopenia or cognitive decline. We examined how mutations impact the duration of "high activity" as a proxy of their health and decline with age. Using the raw movement metric, we quantified the duration and relative time (normalized to each individual's lifespan) of the high activity period across all genotypes (Fig. 3e–f). Average duration of high activity correlated with the average lifespan for each genotype. When normalized by lifespan, both N2 and *daf-16* spend equivalent proportions of their life in the high activity period. However, the *daf-2* mutant spends a significantly lower proportion of its life in the high activity period than the wild-type animals ($p < 0.001$, one-way ANOVA followed by Tukey's HSD test). This decreased behavior has been shown in prior studies and could be attributed to preference for food over exploration and movement, as shown in other *daf-2* alleles[13,28]. Thus, we show that our system recapitulates known behavioral trends on an individual-level with high temporal resolution.

**Effects of DR on longevity and behavior**. We next asked how environmental perturbations influence behavioral decline during the aging process. While DR has been shown to significantly alter aging in *C. elegans*, the interpretation of experimental results can be complex, as it can be difficult to perform precisely controlled DR experiments using conventional closed-culture systems due to depleting food levels over time. To prevent this variability in food levels, we take advantage of the ability of microfluidics to deliver a constant perfusion of bacteria, allowing for precisely controlled food levels over time. To maintain the steady flow of bacteria over long periods of time without the formation of biofilms or bacterial aggregation on the small features of the device, we use our pressure monitoring system, allowing us to culture worms in concentrations up to $OD_{600}$ of 10 (OD 10).

We examined the effects of DR on both the lifespan and healthspan of worms. Unsurprisingly we see a significant decrease in lifespan for worms cultured at OD 10 compared to those at the lower concentration levels ($p < 0.0001$, log-rank, Fig. 4a). The behavior over time was also impacted by DR (Fig. 4, Supplementary Fig. 5). The population average of raw movement across the three food levels had similar decline rates later in life; however, their initial level of movement differed, with worms in higher food concentrations having lower amounts of movement (Fig. 4b–c). This early difference may be attributed to food searching behavior, with worms in DR having increased movement to search for areas with higher food concentrations. The duration and normalized period of high activity movement increases with decreased food concentration (Fig. 4d), with

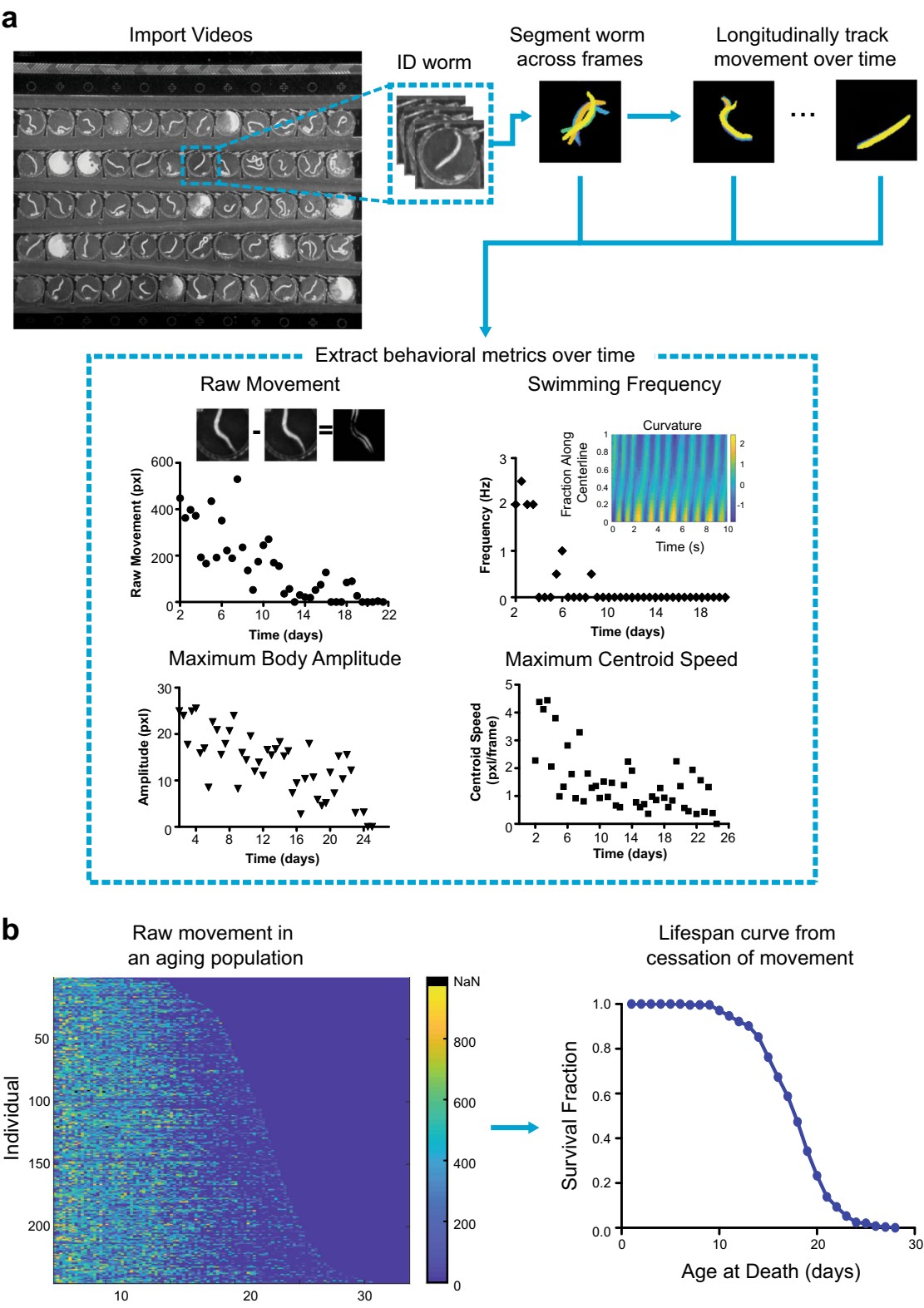

**Fig. 2 Pipeline of behavioral analysis and longevity. a** Overview of the video behavior pipeline. After importing the videos, the code extracts and segments out worms of interest across the set of videos. Behavioral metrics, such as "raw movement", centroid speed, body amplitude, and swimming frequency are then extracted from the segmented videos. **b** Individuals decline of raw movement over time for a population can be used to identify the time of death for each worm based on time of cessation of movement.

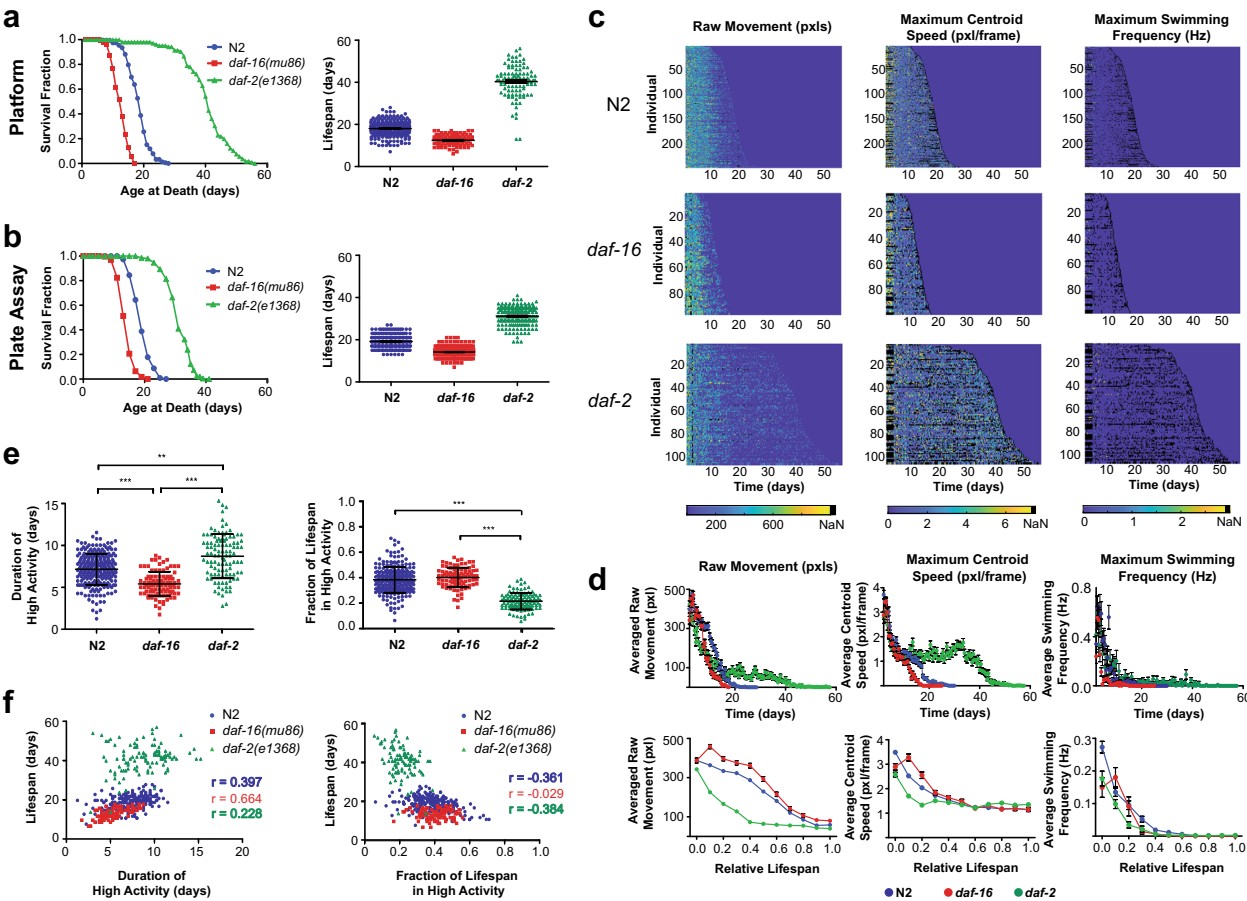

**Fig. 3 Longevity and behavioral decline of IIS mutants. a** Lifespan curves and corresponding dot plot for wild-type N2 (18.00 days ± 0.23, $n = 245$ biologically independent animals), *daf-16(mu86)* (12.49 ± 0.26, $n = 95$ biologically independent animals), and *daf-2(e1368)* (40.32 ± 0.74, $n = 105$ biologically independent animals) cultured in HeALTH. Populations are cultured at 25 °C with food level of $OD_{600}5$. Error is reported as SEM. (log-rank test, $p < 0.0001$). **b** Lifespan curves and dot plot for N2 (19.25 days ± 0.21, $n = 220$ biologically independent animals), *daf-16* (14.18 ± 0.17, $n = 221$ biologically independent animals), and *daf-2* (31.20 ± 0.28, $n = 206$ biologically independent animals) for plate culture. Populations are cultured at 25 °C with food level of $OD_{600}5$. Error is reported as SEM. (log-rank test, $p < 0.0001$). **c** Heatmaps showing individual behavioral decline for different behavioral metrics across different genotypes over time. **d** Population average activity for each metric and relative population average activity for each metric normalized to lifespan. Error bars are SEM. **e** Duration of high activity for raw movement across genotypes and relative fraction of life in high activity for raw movement for each genotype. Error bars are SD. (**$p < 0.01$, ***$p < 0.001$, via one-way ANOVA followed by Tukey's HSD test. Non-significant differences are not marked). **f** Scatter plots of duration of high activity for raw movement vs. individual lifespan and fraction of life with high activity for raw movement vs. individual lifespan. Pearson correlation coefficients are listed for each genotype.

animals at OD 10 having a significantly lower duration of movement, suggesting a possibility of a reduction in healthspan ($p < 0.001$, one-way ANOVA followed by Tukey's HSD test) compared to either OD 5 or OD 2.5 populations.

**Effects of thermal perturbations on longevity and behavior.** Given that *C. elegans* is a poikilotherm and our system can easily vary temperature, we examined the effects of thermal fluctuations on aging. We took inspiration from the natural diurnal temperature cycle *C. elegans* experience in the wild and explored whether dynamic thermal cycling influences their lifespan and healthspan. We measured the lifespan of worms that were switched every 12 h between 15 °C and 20 °C, along with worms cultured at constant temperatures (15 °C, 17.5 °C, and 20 °C) (Fig. 5a). Average lifespan decreases with increased temperature under static conditions, following known trends[9]. Animals undergoing temperature oscillations had an average lifespan similar to worms reared at 20 °C, showing no beneficial extension in lifespan from their time at 15 °C. However, the range of lifespans under the cycling condition was much larger than that of

animals reared at the static 20 °C condition, hinting that variability in temperature contributes to the variability in longevity within a population.

When examining behavior across the constant temperature conditions, all populations appeared to have a high level of activity for approximately half of their lifespan followed by a period of decline (Fig. 5b–c). Populations at lower temperatures had a slight increase in the duration of high activity movement. However, when normalized by lifespan, the proportion of time spent in the high activity period is not significantly different across conditions (Fig. 5d). This suggests temperature has a similar scaling effect on both the period of high activity behavior and lifespan.

Mirroring the trends in lifespan, the population undergoing temperature oscillations had the duration and fraction of life in high activity behavior similar to that of the static 20 °C condition (Fig. 5d). However, when examining averaged raw movement, we observed large fluctuations in average activity (Fig. 5eii), with the magnitude of movement corresponding to the temperature at the time (Fig. 5eiii). Interestingly, under oscillatory conditions, the average movement at 15 °C was reduced compared to movement

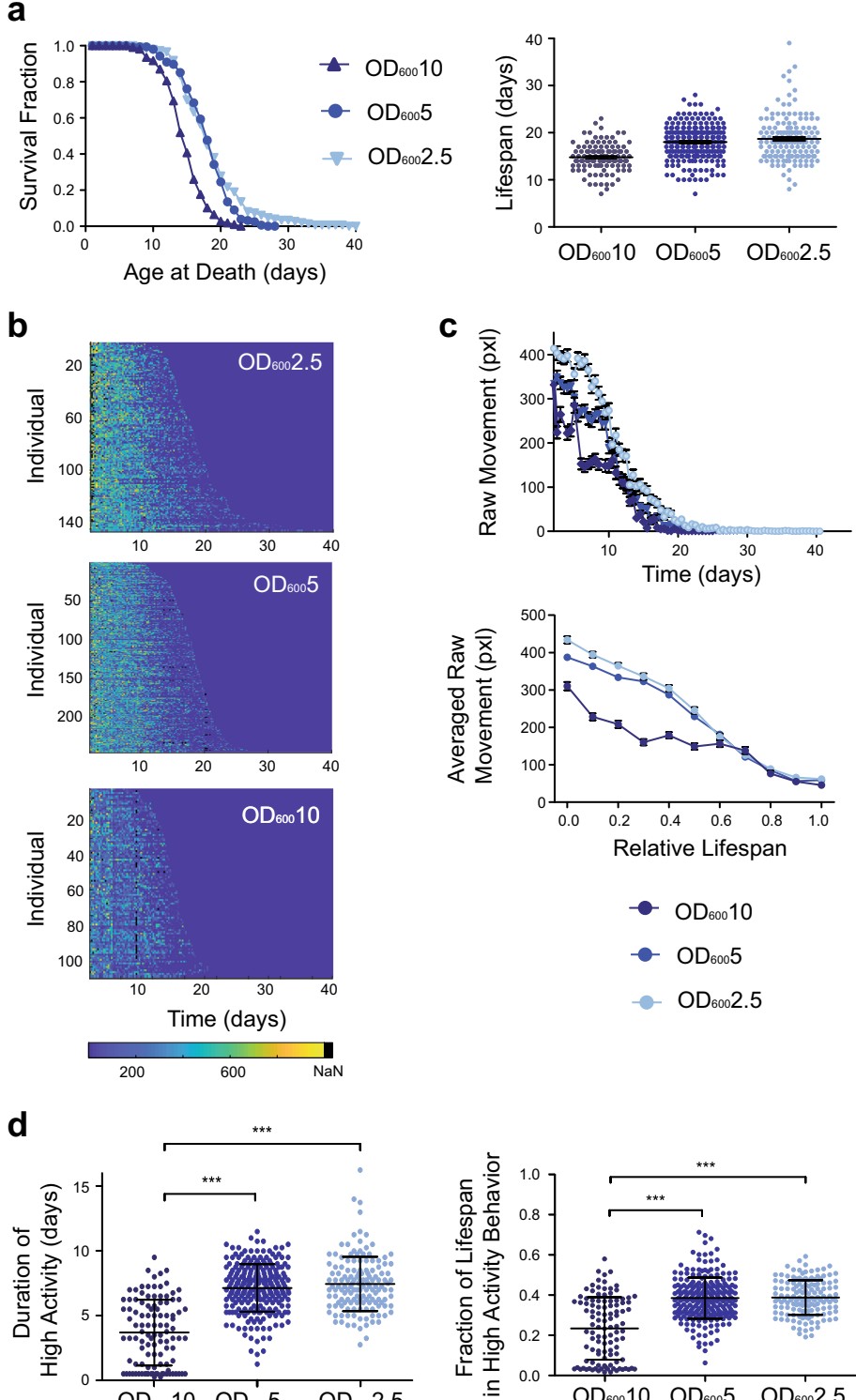

**Fig. 4 Longevity and behavioral decline across different food levels. a** Lifespan curves and corresponding dot plot for wild-type N2 populations cultured at 25 °C at food levels of $OD_{600}2.5$ (18.67 days ± 0.40, $n = 147$ biologically independent animals), $OD_{600}5$ (18.00 ± 0.23, $n =$ biologically independent animals), and $OD_{600}10$ (14.78 ± 0.29, $n =$ biologically independent animals). Error is reported as SEM. (log-rank test, $OD_{600}10$ v. $OD_{600}5/OD_{600}2.5$: $p <$ 0.0001, $OD_{600}5$ v. $OD_{600}2.5$: $p = 0.0854$). **b** Heatmaps showing individual decline in raw movement across food levels over time. **c** Population averaged raw movement and relative fraction of life in high activity for raw movement across food levels. Error bars are SEM. **d** Duration of high activity for raw movement across food levels and relative fraction of life in high activity for raw movement across food levels. Error bars are SD. (***$p < 0.001$, via one-way ANOVA followed by Tukey's HSD test. Non-significant differences are not marked).

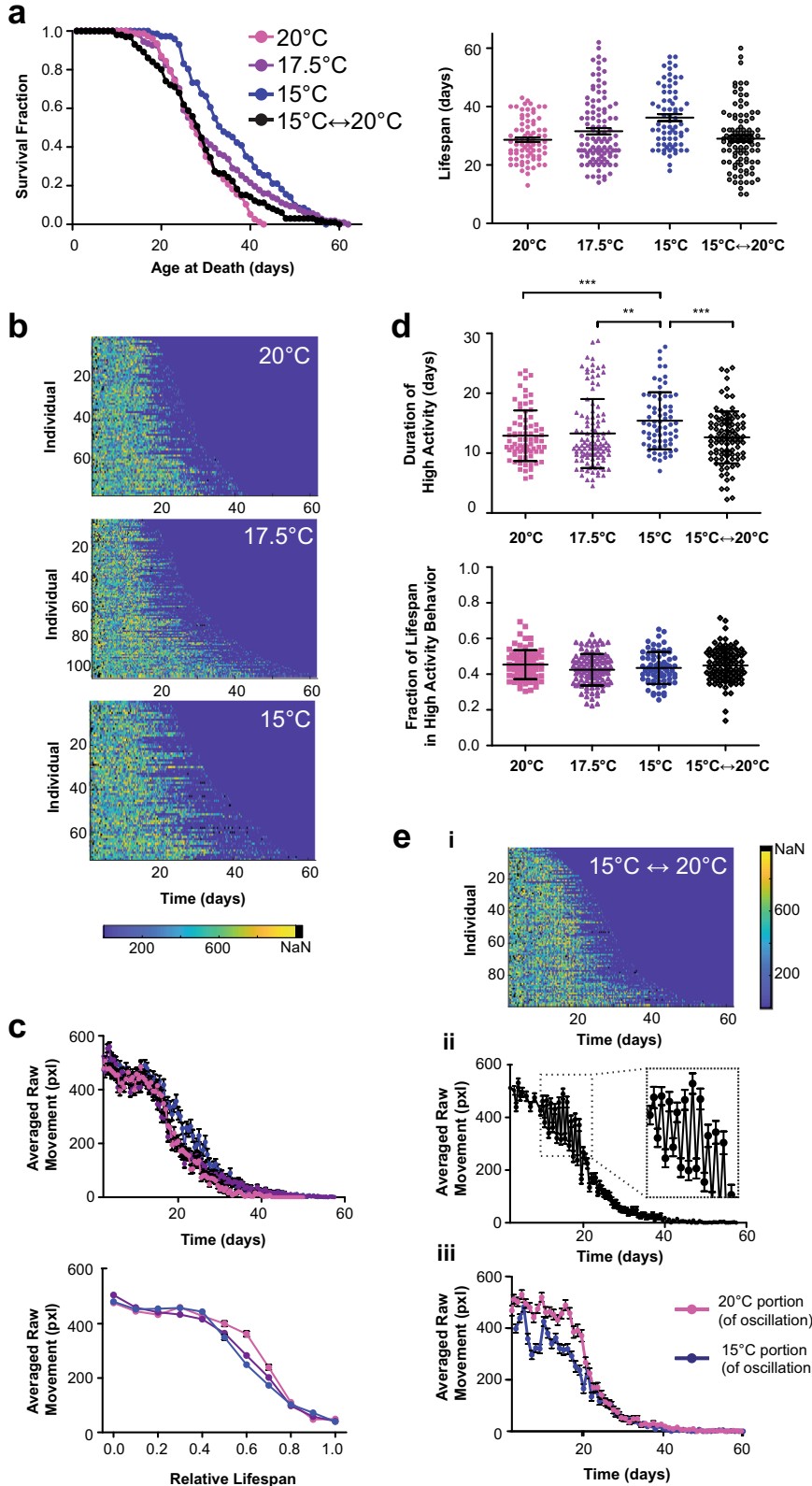

across all other thermal conditions, including the static 15 °C control (Fig. 5c, 5eiii). Due to the ability of our system to obtain frequent behavioral recordings, we were able to subsample a portion of the population with greater temporal resolution and observed a dramatic decrease in movement immediately after the temperature downshift (Supplementary Fig. 6). Over the course of the 12-h cyclical period there was a steady increase in the average

movement approaching the levels observed during the static 15 °C control. Literature has demonstrated the presence of noxious cold receptors in *C. elegans* with an activation threshold around 18 ° C[29]. The drastic decrease in movement could be an initial noxious response to the 15 °C transition, with subsequent increases in movement indicating eventual habituation to the colder thermal condition. This response to downshifted temperatures illustrates

**Fig. 5 Longevity and behavioral decline across thermal perturbations. a** Lifespan curves and corresponding dot plot for wild-type N2 populations cultured at food level $OD_{600}$5 at constant 15 °C (36.21 days ± 1.19, $n = 71$ biologically independent animals), 17.5 °C (31.60 ± 1.09, $n = 107$ biologically independent animals), and 20 °C (28.65 ± 0.80, $n = 77$ biologically independent animals), and under oscillatory temperature conditions, changing from 15 °C to 20 °C every 12 h (29.10 ± 1.04, $n = 99$ biologically independent animals). Error is reported as SEM. (log-rank test, 20 °C v. 17.5 °C $p = 0.0094$, 20 °C v. 15 °C $p < 0.001$, 20 °C v. 15 °C ↔ 20 °C $p = 0.2082$, 17.5 °C v. 15 °C $p = 0.0574$, 17.5 °C v. 15 °C ↔ 20 °C $p = 0.1371$, 15 °C v. 15 °C ↔ 20 °C $p = 0.0004$). **b** Heatmaps showing individual decline in raw movement across static temperature conditions over time. **c** Population averaged raw movement for the static temperature conditions and relative population average activity for raw movement normalized to lifespan across static temperature conditions. Error bars are SEM. **d** Duration of high activity for raw movement across temperatures and relative fraction of life in high activity for raw movement across temperature conditions. Error bars are SD. (**$p < 0.01$, ***$p < 0.001$, via one-way ANOVA followed by Tukey's HSD test. Non-significant differences are not marked). **e** (i) Heatmap showing individual decline in raw movement across oscillatory temperature condition over time. (ii) Population averaged raw movement activity for the oscillatory temperature condition. Error bars are SEM. (iii) Average raw movement in oscillatory condition separated by temperature the population was experiencing at the time. Error bars are SEM.

interesting activity that would otherwise be difficult to obtain with existing methods.

**Combined behavioral metrics for improved lifespan estimates.** With the ability to capture multiple health-related metrics (Fig. 2), we asked if these could be compiled into a single predictive estimator of lifespan across a variety of both genetic and environmental perturbations. We built a LASSO regression model from over 500 different behavioral parameters (Supplementary Note 2) and accurately predicted the lifespan of wild-type N2 population (Supplementary Fig. 7). Using a model developed from N2 data, we examined whether genetic mutants followed the wild-type decline model. When comparing mutants to wild-type, the N2-based model consistently overestimated the lifespan of *daf-16* individuals, while consistently underestimating the lifespan of *daf-2* individuals (Supplementary Fig. 7). This could imply that, unlike the scaling effect of temperature, mutants have fundamentally different behavioral and decline trajectories in aging. Examining combined behavioral metrics could illuminate how behavioral decline due to different perturbations varies from that of wild-type under standard conditions.

**Intrapopulation differences in behavioral decline.** In addition to examining population trends, we examined the effects of stochastic factors on aging in an isogenic population. Similar to previous work, we are able to compare behavioral decline across the shortest- and longest-lived cohorts within a population and demonstrate different decline patterns across cohorts (Fig. 6, Supplementary Fig. 8)[16,17]. As previously reported, the shortest-lived cohort generally exhibited higher levels of activity throughout its relative lifespan compared to the long-lived cohort across all genetic or environmental perturbations. Across thermal conditions, all individuals had similar average decline patterns (Supplementary Fig. 8). Food concentration appears to have a role in the variability of a population's behavioral decline. Shorter-lived cohorts experience higher levels of activity later in life; interestingly, higher food concentrations reduce the differences in decline between the two subpopulations (Fig. 6b, d). This could indicate that at higher food levels, food consumption saturates within the population resulting in reduced differences within the population.

In addition to examining averaged decline within subpopulations, our system enables us to directly track an individual's behavior throughout its relative lifespan. To easily compare individuals throughout their lifespan we performed dimensional reduction, projecting every individual's relative behavioral trajectory into a shared principle component space (Fig. 6e, Supplementary Fig. 9, Supplementary Fig. 10). We directly compare an individual's relative behavioral decline across not only the same population, but also with individuals under

different genetic or environmental perturbations. On a population-level, we see how certain perturbations, such as *daf-2* mutation or high food conditions (OD 10), appear to shift in phenospace, indicating a distinct effect on behavioral decline in comparison to other genetic mutations (*daf-16*) or thermal perturbations. With HeALTH's ability to perform individual behavioral tracking and monitoring, we can observe the spread of the population and examine whether certain perturbations result in increased biological noise for relative behavioral decline. For example, temperature has a more variable behavioral space within the population, indicating increased variability in decline. Furthermore, we can examine the separation between subpopulations (i.e. the short- and long-lived cohorts) and how it varies across perturbations, giving us insight into how stereotyped relative behavioral decline is within a population (Supplementary Fig. 10). For example, *daf-2* shortest- and longest-lived cohorts were separated largely by the first principle component (PC), while *daf-16* relied on the second PC to distinguish between differently aged cohorts. In contrast, wild-type used a combination of the two PCs to separate the subpopulations.

## Discussion

HeALTH allows for automated and robust longitudinal culture of *C. elegans* throughout their lifespan under precise spatiotemporal environmental conditions. The system's inherent modularity and flexibility can accommodate a variety of experimental conditions. It could easily be adapted to monitor different microfluidic devices or environmental perturbations. Furthermore, the overall cost and relatively small physical footprint makes it feasible to create replicates of the system and expand experimental capacity without dramatically increasing space or large equipment requirements. In fact, many of the components used to create the system, such as the main body of the platform, are modified from commercially available parts or DIY kits. Coupled with the design documentation and parts list (detailed in Supplementary Note 1), the system could be recreated in other labs. As a result, it could be used for a wide array of high-throughput aging and longevity studies, providing high-content behavioral data with fine spatial and temporal resolution.

Our system has a unique microfluidic environment, which differs from traditional plate or liquid culture assays. Although worms grown in HeALTH are surrounded by a liquid environment, the geometric 2D confinement experienced on-chip is similar to the 2D environment experienced by those grown on plate. Due to differences from environmental conditions, the quality of reagents, experimental bias, or uncontrollable changes in the environment, it is often challenging to directly compare lifespan durations across different trials and culture conditions[27]. However, to enable future studies and to assess whether the hybrid environment would cause any deleterious effects on lifespan, it was important to ensure that the

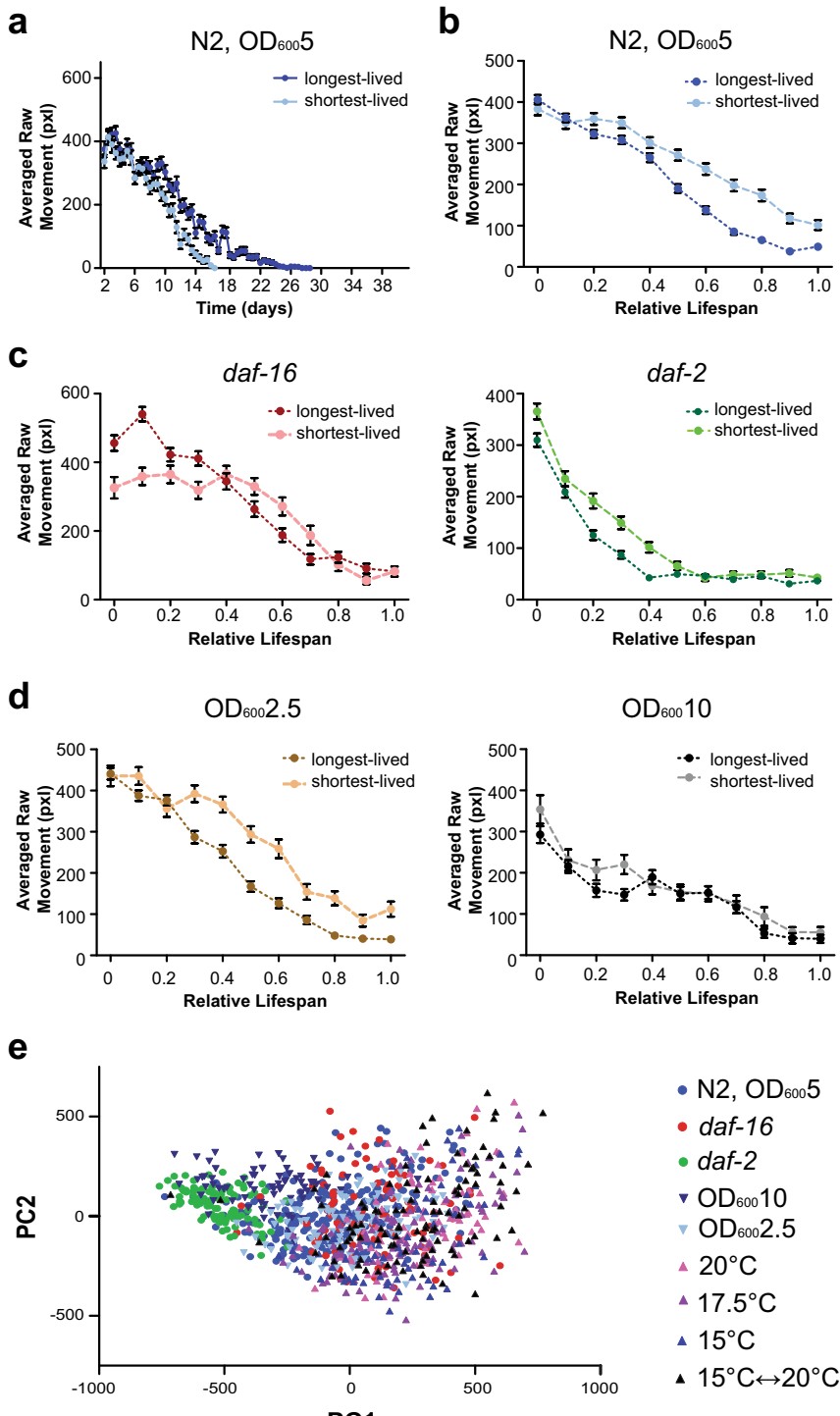

**Fig. 6 Intrapopulation variation across genetic and environmental perturbations. a** Averaged raw movement over time of the short- ($n = 62$ biologically independent animals) and long-lived ($n = 55$ biologically independent animals) subpopulations within the wild-type population culture at 25 °C at $OD_{600}5$ food level. Error bars are SEM. **b** Averaged raw movement over normalized relative lifespan of short- and long-lived WT populations cultured at 25 °C at $OD_{600}5$ food level. Error bars are SEM. **c** Averaged raw movement over the normalized, relative lifespan of the short- and long-lived subpopulations across *daf-16* (shortest-lived: $n = 23$ biologically independent animals, longest-lived: $n = 21$ biologically independent animals) and *daf-2* (shortest-lived: $n = 22$ biologically independent animals, longest-lived: $n = 23$ biologically independent animals) populations at 25 °C at $OD_{600}5$ food level. Error bars are SEM. **d** Averaged raw movement over the normalized, relative lifespan of the short- and long-lived WT subpopulations cultured at 25 °C on $OD_{600}2.5$ (shortest-lived: $n = 45$ biologically independent animals, longest-lived: $n = 37$ biologically independent animals) and $OD_{600}10$ (shortest-lived: $n = 22$ biologically independent animals, longest-lived: $n = 26$ biologically independent animals) food levels. Error bars are SEM. **d** All individuals ($n = 981$ biologically independent animals) plotted in a shared principle component space.

microfluidic environment resulted in no unexpected lifespan trends. We were able to accurately replicate known trends across both genetic and environmental perturbations, thus validating our system.

We demonstrated the power and versatility of HeALTH by examining both population-level and individual decline across different genotypes and environmental perturbations. Controlled diurnal temperature shifts resulted in a wider range of lifespans, suggesting that intra-populational variability is sensitive to environmental perturbations. DR altered the level of activity and relative behavioral decline across populations in a non-linear way. Furthermore, the decreased lifespan for worms cultured at higher bacterial concentrations (OD 10) in HeALTH raises the possibility that existing microfluidic lifespan devices inadvertently enact some form of DR or hormetic stress on their cultured populations, due to the lower bacterial concentrations used by these systems on-chip[19–22]. This in turn could potentially result in confounding effects on longevity. Examining the effects of either DR or controlled diurnal temperature shifts would be extremely difficult, if not impossible, to study on a large-scale with existing methods, but can be performed using HeALTH.

The inherent modularity and flexibility of HeALTH are designed to enable a variety of different assays to examine different facets of the aging process. Due to its liquid culture environment, our system could explore the effects pheromone concentration on behavior and aging. We could also examine the effects of intermittent fasting, perform drug screens, or examine the effect of stressors (such as heat shock, oxidative stress, or osmotic stress). The system may also be applied to behavioral neuroscience, examining how sensory response and cognitive aging change over time. HeALTH could be expanded for any behavioral study that requires precise environmental control with longitudinal tracking. For example, it could be used for immunology or toxicology applications to examine the impact of pathogens or different microbiome interactions at the whole organism scale. By simply rescaling the microfluidic device, this system could be directly used for a variety of small model organisms for similar applications. Furthermore, because we gather high-content behavioral information, our behavioral analysis can be extended from extracting predefined behavioral features to looking at more subtle, less defined behavioral phenotypes. Thus, this platform could be easily adaptable and extendable to a variety of different behavioral assays and aging studies, providing insights into behavior changes as a result of any genetic, environmental, or stochastic perturbations over time.

## Online methods

**C. elegans culture.** The following strains were used: N2, QL12 *daf-16(mu86)* I, and QL390 *daf-2(e1368)* III. The *daf-2(e1368)* III allele is a lifespan-extension mutant without strong deleterious pleotropic effects, such as dauer-like quiescence or Unc like behavior[30,31]. Worms were grown at 20 °C on live *E. coli* OP50 seeded agar plates for two generations. Synchronized L1 larvae of the F2 generation were obtained from a 6-h hatch-off and grown at 20 °C on live *E. coli* OP50 seeded agar plates until reaching the L4 stage, where the worms were either loaded into the microfluidic device or used for manual lifespan assays.

**Microfluidic device fabrication.** Microfluidic devices were fabricated from polydimethylsiloxane using standard soft lithography techniques and bonded to a glass coverslip using plasma bonding. Devices were sterilized by autoclaving before usage.

**E. coli culture/prep.** Bacterial culture protocols are similar to previously published protocols[32,33]. We grew *E. coli* (HB101)

overnight in filtered LB at 37 °C within a shaking incubator. HB101 was chosen to reduce bacterial clumping and aggregation. We then spiked the culture with carbenicillin (final concentration 50 µg/ml) continued culturing for an additional 30 min and chilled it at 4 °C to inhibit growth. We spun down the cultures at 2400 rcf at 4 °C for 20 min and resuspended the pelleted bacteria in S Medium with the surfactant Pluronic F-127 (0.005%), to reduce bacterial aggregation and clumping from occurring within the device, along with carbenicillin (50 µg/ml), and kanamycin (50 µg/ml) to reduce the risk of bacterial contamination during subsequent culture. Bacterial growth was measured by sampling a five-fold dilution of the culture before centrifugation and measuring the $OD_{600}$ of the sample. The samples were resuspended to concentrations of $OD_{600}$ 10 and kept at 4 °C until use. For DR experiments, the $OD_{600}$ 10 stock was diluted with S Medium supplemented with Pluronic, carbenicillin, and kanamycin until reaching the desired concentration.

**Culture on-chip.** Many *C. elegans* aging studies use fluoro-2′-deoxyuridine (FuDR) to prevent progeny from appearing and confounding experimental analysis and lifespan scoring. However, there are several known issues with using FuDR, which result in artificially extended lifespans[34]. As a result, we used C22, a compound that interrupts eggshell formation, creating inviable progeny without impacting the development or longevity of adults[35].

Worms are randomly loaded into the device using a two-step pressurized loading process previously developed in our lab[26]. They are then mounted onto a temperature control module and cultured at 20 °C in bacterial concentrations of $OD_{600}$ 10 supplemented with 5uM C22 on-chip to mitigate any effects DR may have on the developmental process and prevent progeny. When mounting devices on the temperature control module, a thin layer of silicon oil was applied between the silicon surface and the glass coverslip for enhanced thermal conductivity and improved contrast for imaging. At Day 2 of adulthood, worms were then shifted to the desired temperature and food level. Every other day, upstream in-line filters were changed to prevent excessive bacterial accumulation. Every four days, we exchanged upstream and downstream tubing and downstream resistance devices to prevent bacterial accumulation and added new bacteria at the desired concentration supplemented with 5uM C22 to the pressurized food source. All filters, tubing, and food source bottles were sterilized by autoclaving before use to prevent potential contamination.

Custom reservoirs filled with the bacteria are pressurized by the pressure box to creating flow through the device. A 0.22um filter is placed between the pressure source and reservoir to reduce the risk of contamination. To prevent bacterial accumulation, the bacteria within the pressurized reservoirs is continuously agitated with a stir bar. Additionally, a series of sterile mesh filters are placed between the reservoir and device to prevent any bacterial aggregates from entering and clogging the microfluidic device. The resistance device downstream of the device and pressure sensor limits the flow rate of the device. We used an average flow rate of approximately 15 µL/min across all conditions. Every hour, devices underwent an increased "pulsed" flow for 5 s, with an average flow rate of approximately 275 µL/min, to provide additional mechanical stimulation and to help reduce bacterial accumulation within the upstream tubing.

**Manual lifespan assays.** Manual lifespan assay protocols are similar to previously published work[32,33]. Six centimeters NGM plates were supplemented with 5uM C22 and 50ug/ml of both carbenicillin and kanamycin to prevent progeny and potential

contamination. For the first two days, worms were placed on plates seeded with lawns of 225 μl of HB101 concentrated at OD$_{600}$ 10 and cultured at 20 °C. At Day 2 of adulthood worms were transferred to the desired environmental conditions and subsequently transferred regularly to fresh plates until the end of the reproductive period. Worms were gently stimulated with a platinum wire pick every other day to assay for movement; lack of response was scored as death. We performed at least three different biological replicates for each genotype examined in the manual assays. Raw lifespan data for the plate assays are included in SI.

**Video acquisition**. Videos were taken twice an hour for 10 s at an acquisition rate of 14 fps using a 1.3 Megapixel monochrome CMOS camera (Thorlabs DCC1545M camera) coupled with a 10X close focus zoom lens (Edmund #54-363). The magnification was set to allow all chambers to be within the field of view (24 mm × 19.2 mm). Illumination was provided by a set of concentric red LED rings (Super Bright LEDs 60 and 80 mm LED Halo Headlight Accent Lights) to reduce the amount of blue light exposed to the worm. To prevent excessive, strong light exposure to the worm, the LED was on only while viewing devices or during video recordings.

**Behavioral analysis**. Throughout the lifespan of each individual worm across the genetic and environmental conditions, we longitudinally tracked and measured its behavior over time. Due to the large volume and frequency of behavioral recordings, we initially examined and analyzed only a subset of our data, looking at behavior every 12 h. For each time point, we sampled a video of worms undergoing a constant flow rate and a video of worms immediately after experiencing an increased, "pulsed" flow. When examining behavior immediately after switching to 15 °C for the thermal oscillatory case, we examined videos every other hour for increased temporal resolution.

A major behavioral characteristic we examined was the duration of high activity movement. We defined "high activity" by examining the compiled movement data of the experimental condition of interest. For metrics with more defined separation between high and low activity values—such as raw movement, frequency, and centroid speed—we performed k-means clustering to identify two separate populations. For metrics with more subtle decline patterns, such as body amplitude, we tracked the moving average of each individual and identified the point where it fell below the overall average movement of the individual. Activity before that point was classified has high activity, whereas movement after was characterized as low activity.

To examine intrapopulation differences we divided each experimental condition into cohorts based on their lifespan. The shortest-lived cohort consisted of the bottom 20th percentile, while the longest-lived cohort consisted of the top 20th percentile for each experimental condition. Relative behavioral decline within experimental conditions was compared for the shortest and longest-lived cohorts, using the Kolmogorov–Smirnov test; no significant difference was found. The MATLAB code for identifying high activity movement and for examining intrapopulation differences is available on GitHub[36].

To perform principle component analysis we used the built-in principle component analysis of raw data function in MATLAB. We examined the average relative raw movement for each individual across all conditions, resulting in a 1054 × 11 matrix (with 1054 individuals and 11 binned relative time periods).

**Predictive modeling**. Due to the large volume of behavioral metrics we extract and the large prevalence of collinearity across the features, we chose a LASSO regression model with 10-fold cross validation to create a single predictive estimate of lifespan. We used a variety of different predictive parameters within the model, such as the maximum daily value or the daily average value, or the rate of decline. When creating the model we censored very sparse data (largely caused by unreliable segmentation) and performed basic feature imputation using the median of each category on remaining missing values. Supplementary Note 2 lists the parameters inputted into the model, along with the model selected variables. The code is freely available and uploaded to GitHub[36].

**Statistics and reproducibility**. For each experimental condition, we performed at least three independent biological replicates. Sample sizes for each condition are included within the figure legends and are comparable to traditional lifespan experiment sample sizes. Individuals were censored in cases where multiple individuals were loaded within the same chamber, or if the individual experienced a premature death not caused by aging, such as bursting (i.e. intestinal extrusion through the vulva). Raw lifespan data for each analyzed individual, along with corresponding experimental conditions and trial information, is included in SI. To identify and characterize the sources of variance across experimental trials we performed a mix-model approach. Similar to previously published work[27] we used GLMs using the *lme4* v.1.12 package in R. We accounted for variance due to different trials, and devices or plates within the trial; however, we did not include variance due to different batches of bacterial food.

Lifespans for both on-chip and plate experiments were analyzed via Kaplan–Meier and significance was calculated with the log-rank test in JMP Pro14. Two-sample Kolmogorov–Smirnov tests were performed in MATLAB (2018b) using a built-in function. Remaining statistical tests (such as one-way ANOVA followed by Tukey's HSD test, Pearson's correlation coefficient) were performed in GraphPad Prism 5.

**Reporting summary**. Further information on research design is available in the Nature Research Reporting Summary linked to this article.

## Data availability
Raw lifespan data for all individuals is included as Supplementary Data 1. Behavioral information for each subpopulation is also included as Supplementary Data 2. Raw behavioral videos are available from the authors upon reasonable request.

## Code availability
A custom LabVIEW (2013) code was created to obtain and record the behavioral videos. Behavioral analysis of the video data was performed with a custom MATLAB (2018b) script. The code for both the user GUI and the behavioral analysis pipeline is available on GitHub (https://github.com/kim-le63/HeALTH_Tracker/)[36].

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

## Acknowledgements

The authors would like to thank K. Broderick for his work on the analysis pipeline, along with L. Zhang and M. Green for help with experimental preparatory work. This work was supported by NIH R01AG056436, NIH R01GM088333, R01AG035317, and NSF DGE-1650044.

## Author contributions

M.Z. and H.L conceived the project. M.Z., Y.C., D.S.P., and K.L. developed the HeALTH platform and culturing protocols. K.L. conducted the experiments and analyzed the data in this paper; M.Z. and Y.C. conducted preliminary experiments. J.W. and D.S.P. assisted in conducting the experiments and plate controls. Y.C. and K.L. developed the analysis pipeline. All authors wrote or edited the manuscript.

## Competing interests

The authors declare no competing interests.
