## [Peer Review File · Communications Biology]

Reviewers' comments:

Reviewer #1 (Remarks to the Author):

In the paper "HeALTH: An Automated Platform for Long-term Longitudinal Studies of Whole Organisms under Precise Environmental Control", Le et al develop a novel microfluidic device called Health and Lifespan Testing Hub (HeALTH) for behavioral monitoring in a long-term and controlled manner. They demonstrate that this device can precisely control the worm's environment longitudinally, allowing long-term lifespan and healthspan experiments to be performed. In addition, they show that the HeALTH system can reproduce the behavior and longevity phenotypes previously observed in dietary-restricted worms, worms under thermal perturbations, and IIS mutants. Finally, they used data collected in this device to build a model for better estimating lifespan and healthspan in different worm populations.

This is a comprehensive study of a potentially useful device that will be of interest to the field. Overall, the data are of high quality and support the author's conclusions; they use the right controls, adequate numbers of worms and proper statistical analysis in their experiments. The rationale and experimental design are clearly presented in the paper. I have only four minor comments that, if addressed, could provide better clarity for the manuscript:

- 1) Since worms grown in liquid are phenotypically and metabolically different from those grown on solid plates, authors should have emphasized their method is more like liquid culture rather than plate culture.
- 2) For Figure 2b, it would be clearer to have a side-by-side manually performed control lifespan experiment (in liquid culture) to show how the results differ between the manually performed assay and the new HeALTH assay.
- 3) For Figures 3 and 4, state lifespan assay conditions in the figure legend for clarification. A careful look at the data reveals that the N2 lifespan in Figure 3a mimics the N2 lifespan in Figure 4a with the OD600-5 condition. The authors should explain the reason for that difference.
- 4) Data in Figure 4 is interpreted in this statement: "animals at OD600-10 having a significantly lower duration of movement, suggesting a shorter healthspan". Considering that there is more food available at OD600-10, the authors cannot rule out the possibility that the worms just spend less time finding food and that this change in movement has nothing to do with healthspan.

Reviewer #2 (Remarks to the Author):

The manuscript presents an automated system for conducting *C. elegans* lifespan and behavioral experiments. Initially, the authors discuss the need for studying the role of environmental factors in organismal health lifespan and present issues with the currently available methods for performing these studies in the model organism *C. elegans* (lack of precise spatiotemporal control, scalability, experimental robustness). Subsequently, they introduce HeALTH, their own system, which is capable of precise spatiotemporal temperature control, consistent supply of food (and possibly drug compounds) of specific concentration, and high throughput acquisition of longitudinal data at the individual worm level. They then provide details on the various components of the device (video acquisition, temperature control, microfluidic system for consistent food and drug supply, clog prevention sensor, software for device control, data generation and analysis) and explain the types of behavioral data that the system can track longitudinally during aging. The authors used worm strains with well-documented lifespans (N2, *daf-2(e1368)* and *daf-16(mu86)*) to demonstrate that their system generates data that matches traditional manual experiments. They also utilized their data to assess the healthspan of these worms, and point out that long-lived mutants spent a smaller portion of their life having high activity compared to short-lived and wild type animals. The authors also performed experiments utilizing different worm food (bacteria) concentrations, showing a reverse relation between lifespan and bacterial abundance that is in line with the paradigm of dietary restriction. They point out that worms that were provided with high food concentrations have a shorter high activity period. In addition, the authors examined the effects of temperature on worm lifespan, examining both consistent temperatures (15°, 17.5°, and 20°) and an alternation of 15° and 20° that emulates worm living conditions in the wild. Worms that lived in these alternating conditions have the same average lifespan as

worms that lived consistently at 20° but exhibit higher variance. Lower temperatures increased worm lifespan and the high activity period in a proportionate manner. Beyond these experiments, the authors utilized the individual worm metrics that the system collects to build a LASSO regression model and identify behavioral parameters that can accurately predict worm lifespan. They were successful in creating a model that can predict the lifespan of N2 worms at 25°. Unfortunately that model lacks accuracy in regard to long- or short- lived mutants. Finally, the authors used individual worm data to analyze differences between the longest and shortest cohorts of each experimental population. The results suggest that shorter-lived cohorts exhibit a longer high activity period relative to their total lifespan, and that this effect is lost in high food concentrations. They also performed principal component analysis analyzing the effects of genotype, food abundance and temperature.

All in all, even though the ideas of automated *C. elegans* lifespan, individual longitudinal tracking, or precise spatiotemporal environmental control are not novel, and there are already devices that can achieve some of these goals, the system presented in this manuscript represents a considerable improvement in both its technical implementation and combination of features. As far as the scientific impact is concerned, as the authors demonstrate with their attempts at LASSO regression or sub-population cohort analysis, having access to high volumes of individual animal data can allow a researcher to address lifespan and healthspan at a level far beyond what is feasible with traditional means. There are a few remaining issues that need to be addressed: Some of the lifespan data shown does not adhere to expectations. The lifespans shown at figure S3 and those represented at figure S7 are long for 25° conditions. The authors should add an extra column at the final table of supplemental note 2, indicating what actual behavioral metrics their selected variables represent. The current format ('ampData_maxVal_avgValDay_Day0') does not provide this information clearly. The authors should mention the initial behavioral dimensions they used for principal component analysis.

What is the purpose of Pluronic F-127 in the bacterial (HB101) cultures?

Reviewer #3 (Remarks to the Author):

HeALTH: An Automated Platform for Long-term Longitudinal Studies
of Whole Organisms under Precise Environmental Control

This manuscript develops a new platform to study animals long-term and in a longitudinal level manner. Overall, it is unclear how interesting this will be to a general audience. The platform has limited uses. Other than movement, it is unclear what can be examined. In addition, it is not clear why the *daf-2* allele shows a difference in lifespan on plate vs platform, there is no mention of cost or feasibility for other labs and there is limited/variable results comparing plate vs platform. Therefore, I would reject the manuscript.

Here are some specific questions to address when submitting to a more specialized journal.

Why is this of interest to the general audience?

Why was *daf-2(e1368)* used? What is the reason for the large discrepancy in the lifespan on the plate versus this platform?

How feasible is this for other labs?

Although mentioned, it's difficult to understand how this will be done. "We could also examine the effects of intermittent fasting, perform drug screens, or examine the effect of stressors (such as heat shock, oxidative stress, or osmotic stress)"

We thank the editor and reviewers for their comments and feedback. We have responded to all comments and questions. Below is our response along with references to changes made in the manuscript.

Reviewers' comments:

Reviewer #1 (Remarks to the Author):

In the paper “HeALTH: An Automated Platform for Long-term Longitudinal Studies of Whole Organisms under Precise Environmental Control”, Le et al develop a novel microfluidic device called Health and Lifespan Testing Hub (HeALTH) for behavioral monitoring in a long-term and controlled manner. They demonstrate that this device can precisely control the worm’s environment longitudinally, allowing long-term lifespan and healthspan experiments to be performed. In addition, they show that the HeALTH system can reproduce the behavior and longevity phenotypes previously observed in dietary-restricted worms, worms under thermal perturbations, and IIS mutants. Finally, they used data collected in this device to build a model for better estimating lifespan and healthspan in different worm populations.

This is a comprehensive study of a potentially useful device that will be of interest to the field. Overall, the data are of high quality and support the author’s conclusions; they use the right controls, adequate numbers of worms and proper statistical analysis in their experiments. The rationale and experimental design are clearly presented in the paper. I have only four minor comments that, if addressed, could provide better clarity for the manuscript:

1) Since worms grown in liquid are phenotypically and metabolically different from those grown on solid plates, authors should have emphasized their method is more like liquid culture rather than plate culture.

We thank the reviewer for their comments. We agree that there are phenotypic differences between worms grown in liquid culture and on solid plates, which can result in differences in lifespan. Although worms grown in HeALTH are surrounded by a liquid environment, the geometric 2D confinement experienced on-chip creates a physical environment that differs from liquid culture and more closely resembles plate conditions. For example, as the worm grows, its diameter approaches the height of the chamber (80µm). This results in a confinement effect that limits its range of motion in the Z-axis, comparable to the surface tension experienced on the surface of a solid agar plate. As a result, the worms maintained in our platform experience a hybrid of solid and liquid culture environments. See **Supplemental Videos 1-15** for examples of how the worms transition from swimming behavior, common to liquid, to more crawling-like behavior caused by the confinement on-chip. We have also included below time-lapse images of worms moving 1) on plate, 2) in buffer, and 3) on-chip when the geometric confinement begins to occur due to the size of the worm (Fig. 1).

Figure 1. Time-lapse images of behavior across environments. a) Crawling behavior on a solid agar plate (Day 1 Adult). b) Swimming behavior in buffer (Day 1 Adult). c-e) Behavior on-chip (from a Day 3, Day 4, and Day 6 Adult)

In addition, our worms are actually cultured on plates until the L4 larval stage before being loaded into the HeALTH platform, so nearly all of their larval development is on a solid surface, unlike many liquid culture lifespans in which all stages of development occur in liquid.

Thus, we think that neither the traditional plate culture nor liquid culture is the perfect control for HeALTH, and that HeALTH is a hybrid culture environment. We use conventional plate culture as a reference to ensure that biological and chemical reagents perform as expected. We have updated the text to reflect this (lines 169-171, 320-326). The updated text is also included below.

“This ability to recapitulate both the trends and variability in comparison to traditional plate assays underscores the viability of using HeALTH to perform lifespan studies.” (lines 169-171)

“Our system has a unique microfluidic environment, which differs from traditional plate or liquid culture assays. Although worms grown in HeALTH are surrounded by a liquid environment, the geometric 2D confinement experienced on-chip is similar to those grown on plate. To enable future studies, and to assess whether the hybrid environment would cause any deleterious effects on lifespan, it was important to ensure that the microfluidic environment resulted in no unexpected lifespan trends. We were able to accurately replicate known trends across both genetic and environmental perturbations, thus validating our system.” (lines 320-326)

2) For Figure 2b, it would be clearer to have a side-by-side manually performed control lifespan experiment (in liquid culture) to show how the results differ between the manually performed assay and the new HeALTH assay.

We appreciate the reviewer's recommendation. As discussed above, we believe that the lifespans of the worms cultured in HeALTH are not directly comparable to either the solid plate or liquid culture, due to the various environmental differences across the conditions. We have shown that the lifespans are within **range** and preserve the **trends** observed in literature, even to those done in liquid culture conditions [Bishop and Guarente, *Nature*, 2007, **447**, 545-549 and Win *et al.*, *Aging Dis.*, 2013, **4**, 178-185]. We are not interested in comparing the absolute values of lifespans. We have updated the manuscript to reflect this point (lines 164-166, 169-171). The updated text is shown below. We do, however, demonstrate that worms grown in HeALTH have lifespan **trends** that follow our own parallel plate culture, which ensures that the organisms and reagents are behaving as expected. As a result, we would not expect any additional insights gained from a parallel liquid culture.

“Longevity assays performed in HeALTH are comparable to lifespan trends found using traditional plate assay, for both wild-type and mutant populations (**Fig. 3a**).” (lines 164-166)

“This ability to recapitulate both the trends and variability in comparison to traditional plate assays underscores the viability of using HeALTH to perform lifespan studies.” (lines 169-171)

3) For Figures 3 and 4, state lifespan assay conditions in the figure legend for clarification. A careful look at the data reveals that the N2 lifespan in Figure 3a mimics the N2 lifespan in Figure 4a with the OD600-5 condition. The authors should explain the reason for that difference.

We thank the reviewer for the suggestion and have modified the figure legends accordingly for greater clarity (lines 620-624, 636-638). The N2 lifespan used as a control in both Figure 3a and Figure 4a are from same experimental data (25°C under OD600-5 food levels) and thus have the same lifespan.

“Populations are cultured at 25°C with food level of OD₆₀₀5 (error bars are SEM, $p < 0.0001$, via log-rank test). (*bottom*) Lifespan curves and bar graph for N2 (19.25 ± 0.21 , $n = 220$), *daf-16* (14.18 ± 0.17 , $n = 221$), and *daf-2* (31.20 ± 0.28 , $n = 206$) for corresponding plate conditions. Populations are cultured at 25°C with food level of OD₆₀₀5 (error bars are SEM, $p < 0.0001$, via log-rank test).” (lines 620-624)

“Lifespan curves for wild-type N2 populations cultured at 25°C at food levels of OD₆₀₀2.5 (18.67 ± 0.40 , $n=147$ individuals), OD₆₀₀5 (18.00 ± 0.23 , $n=245$), and OD₆₀₀10 (14.78 ± 0.29 , $n=108$).” (lines 636-638)

4) Data in Figure 4 is interpreted in this statement: “animals at OD600-10 having a significantly lower duration of movement, suggesting a shorter healthspan”. Considering that there is more food available at OD600-10, the authors cannot rule out the possibility that the worms just spend less time finding food and that this change in movement has nothing to do with healthspan.

We appreciate the reviewer's comment; we agree that there is a possibility that the decreased period of high levels of movement is due to the reduced amount of time spent searching for food, although at all food levels the odorants and tastants from food are all present. We have modified the manuscript in lines 218-221:

“The duration and normalized period of high activity movement increases with decreased food concentration (**Fig. 4c**), with animals at OD 10 having a significantly lower duration of

movement, suggesting a possibility of a reduction in healthspan ($p < 0.0001$, one-way ANOVA followed by Tukey's HSD test) compared to either OD 5 or OD 2.5 populations."

Reviewer #2 (Remarks to the Author):

The manuscript presents an automated system for conducting *C. elegans* lifespan and behavioral experiments. Initially, the authors discuss the need for studying the role of environmental factors in organismal health lifespan and present issues with the currently available methods for performing these studies in the model organism *C. elegans* (lack of precise spatiotemporal control, scalability, experimental robustness). Subsequently, they introduce HeALTH, their own system, which is capable of precise spatiotemporal temperature control, consistent supply of food (and possibly drug compounds) of specific concentration, and high throughput acquisition of longitudinal data at the individual worm level. They then provide details on the various components of the device (video acquisition, temperature control, microfluidic system for consistent food and drug supply, clog prevention sensor, software for device control, data generation and analysis) and explain the types of behavioral data that the system can track longitudinally during aging.

The authors used worm strains with well-documented lifespans (N2, *daf-2(e1368)* and *daf-16(mu86)*) to demonstrate that their system generates data that matches traditional manual experiments. They also utilized their data to assess the healthspan of these worms, and point out that long-lived mutants spent a smaller portion of their life having high activity compared to short-lived and wild type animals. The authors also performed experiments utilizing different worm food (bacteria) concentrations, showing a reverse relation between lifespan and bacterial abundance that is in line with the paradigm of dietary restriction. They point out that worms that were provided with high food concentrations have a shorter high activity period. In addition, the authors examined the effects of temperature on worm lifespan, examining both consistent temperatures (15°, 17.5°, and 20°) and an alternation of 15° and 20° that emulates worm living conditions in the wild. Worms that lived in these alternating conditions have the same average lifespan as worms that lived consistently at 20° but exhibit higher variance. Lower temperatures increased worm lifespan and the high activity period in a proportionate manner.

Beyond these experiments, the authors utilized the individual worm metrics that the system collects to build a LASSO regression model and identify behavioral parameters that can accurately predict worm lifespan. They were successful in creating a model that can predict the lifespan of N2 worms at 25°. Unfortunately that model lacks accuracy in regard to long- or short- lived mutants. Finally, the authors used individual worm data to analyze differences between the longest and shortest cohorts of each experimental population. The results suggest that shorter-lived cohorts exhibit a longer high activity period relative to their total lifespan, and that this effect is lost in high food concentrations. They also performed principal component analysis analyzing the effects of genotype, food abundance and temperature.

All in all, even though the ideas of automated *C. elegans* lifespan, individual longitudinal tracking, or precise spatiotemporal environmental control are not novel, and there are already devices that can achieve some of these goals, the system presented in this manuscript represents a considerable improvement in both its technical implementation and combination of features. As far as the scientific impact is concerned, as the authors demonstrate with their attempts at LASSO regression or sub-

population cohort analysis, having access to high volumes of individual animal data can allow a researcher to address lifespan and healthspan at a level far beyond what is feasible with traditional means. There are a few remaining issues that need to be addressed:

1) Some of the lifespan data shown does not adhere to expectations. The lifespans shown at figure S3 and those represented at figure S7 are long for 25° conditions.

We appreciate the reviewer's comments on the work. In terms of the lifespans obtained at 25°C conditions, we acknowledge that our lifespans are longer than existing reported lifespans of N2 individuals.

The lifespan data obtained via traditional plate assay controls had comparable average lifespan compared to those obtained via the HeALTH platform (19.25 ± 0.21 vs. 18.00 ± 0.23 days for N2 animals). This indicates that the lifespan extension was not an artifact due to the HeALTH system but is self-consistent within the lab. This extension (both on-chip and within our plate control assays) could be due to wide range of reasons. For example, in terms of the impact of food quality, the nutrient rich HB101 [Shtonda and Avery, *J. Exp. Biol.*, 2006, **209**, 89-102] has been shown to result in extended lifespan compared to the commonly used OP50 *E. coli* strain [Soukas *et al.*, *Genes Dev.*, 2009, **23**, 496-511 and Chen *et al.*, *Aging Cell*, 2013, **12**, 932-940]. Unlike many lifespan assays, we use non-growing bacteria (treated with antibiotics), which has been shown to prevent premature death due to bacterial colonization in wild-type animals and result in longer lifespans [Podshivalova *et al.*, *Cell Rep.*, 2017, **19**, 441-450]. Additionally, the concentration of food used in our experiments (OD_{6005}) is much lower than the *ad libitum* levels commonly used in lifespan experiments [Sutphin and Kaeberlin, *J. Vis. Exp.*, 2009, **27**, 1152], resulting in extended lifespan due to dietary restriction. Furthermore, lifespan assays are known to be highly variable across studies, due to differences ranging from experimental conditions, the quality of reagents, experimenter bias, or uncontrollable changes in the environment [Lucanic *et al.*, *Nat. Commun.*, 2017, **8**, 14256]. As a result, it is difficult to directly compare lifespan durations across different labs and experiments.

2) The authors should add an extra column at the final table of supplemental note 2, indicating what actual behavioral metrics their selected variables represent. The current format ('ampData_maxVal_avgValDay_Day0') does not provide this information clearly.

We thank the reviewer for the suggestion and have made the requested change to the table within Supplemental Note 2.

3) The authors should mention the initial behavioral dimensions they used for principal component analysis.

We thank the reviewer for the recommendation and have added an additional section within the Methods section (under Behavioral Analysis) detailing the PCA initial dimensions on lines 470-473 (see below for revised text).

“To perform principle component analysis we used the built-in principle component analysis of raw data function in MATLAB. We examined the average relative raw movement for each individual across all conditions, resulting in a 1054 x 11 matrix (with 1054 individuals and 11 binned relative time periods).”

4) What is the purpose of Pluronic F-127 in the bacterial (HB101) cultures?

We appreciate the reviewer's note. The Pluronic F-127 is used as a surfactant to reduce bacterial aggregation and clumping from occurring within the media and subsequently flowing into the device. This has been clarified in the methods section on lines 375-378 (see below for revised text).

“We spun down the cultures at 2400 rpm at 4°C for 20 minutes and re-suspended the pelleted bacteria in S Medium with the surfactant Pluronic F-127 (0.005%), to reduce bacterial aggregation and clumping from occurring within the device, along with carbenicillin (50 µg/ml), and kanamycin (50 µg/ml) to reduce the risk of bacterial contamination during subsequent culture.”

Reviewer #3 (Remarks to the Author):

HeALTH: An Automated Platform for Long-term Longitudinal Studies
of Whole Organisms under Precise Environmental Control

This manuscript develops a new platform to study animals long-term and in a longitudinal level manner. Overall, it is unclear how interesting this will be to a general audience. The platform has limited uses. Other than movement, it is unclear what can be examined. In addition, it is not clear why the *daf-2* allele shows a difference in lifespan on plate vs platform, there is no mention of cost or feasibility for other labs and there is limited/variable results comparing plate vs platform. Therefore, I would reject the manuscript.

Here are some specific questions to address when submitting to a more specialized journal.

1) Why is this of interest to the general audience?

We appreciate the reviewer's comment; we respectfully disagree. Due to the increased prevalence of age-related diseases, there is a significant need to understand aging to extend the period of healthy living. Environmental factors have been shown to influence the aging process. However, it can be difficult to study these effects in a controlled manner. Conventional methods often do not provide precise spatiotemporal environmental control, lack robustness, or require significant amounts of manual labor making it difficult to scale. This system allows researchers to perform large-scale lifespan and aging screens under precise, environmental perturbations and observe how movement and behavior changes with age. This will allow researchers to gain powerful insights into how environmental factors influence health and lifespan. Furthermore, due to the longitudinal behavioral tracking of individuals this system enables researchers to observe how variability during the aging process is altered across different genetic or environmental perturbations, giving insight into the role of stochasticity in the aging process.

Though the manuscript describes how specific environmental perturbations influence the aging process, we also discuss how the system can be used beyond the aging applications (lines 339-353). Due to the use of microfluidics and its ability to perform long-term behavioral monitoring, the HeALTH system could be used to perform large scale drug screens, study cognitive behavioral decline in response to sensory cues,

or observe how environmental stressors, such as heat shock or oxidative stress, influence health and lifespan. Its ability to provide specific environmental cues with high spatiotemporal resolution allows users to understand how temporal cues and dynamic perturbations can influence behavior and aging. Additionally, it is easily adapted to short-term behavioral assays, such as examining the impact of pathogens, microbiome interactions, or sleep making it of wide relevance and interest to those in neuroscience, toxicology, and immunology, as well as those in the aging community.

2) Why was *daf-2(e1368)* used? What is the reason for the large discrepancy in the lifespan on the plate versus this platform?

The *daf-2(e1368)* allele is a well-known and widely used lifespan-extension mutant [Kimura *et al.*, *Science*, 1997, **277**, 942-946, Gems *et al.*, *Genetics*, 1998, **150**, 129-155, and Martinez *et al.*, *eLife*, 2020, **9**]. Other *daf-2* mutants, such as *daf-2(e1370)*, have deleterious pleiotropic effects, such as the strong dauer-like quiescence and Unc like adult behavior reported with the *daf-2(e1370)* allele [Gems *et al.*, *Genetics*, 1998, **150**, 129-155].

In the literature, it is known that it is difficult to meaningfully directly compare the exact lifespan durations from one culture method to another. *Environmental differences across the systems could have genotype-specific impacts on lifespan*. For instance, under dietary restriction, *daf-2* has a larger extension in mean lifespan compared to the increases seen in the N2 wild-type and *daf-16* strain [Bishop and Guarente, *Nature*, 2007, **447**, 545-549]. This could be occurring on-chip under the culture conditions shown in Figure 3a (OD₆₀₀5), which was later shown (in Figure 4a) to extend lifespan due to dietary restriction. In contrast, due to evaporation of the liquid media within the seeded bacterial lawn, the effective bacterial concentration on plate is likely higher than that experienced on-chip. As a result, the worms on plate could have experienced higher levels of food without experiencing dietary restriction. However, the preservation of lifespan trends across the different culture methods validates the ability of HeALTH to serve as a useful tool for examining how perturbations influence lifespan.

3) How feasible is this for other labs?

We appreciate the reviewer's comment. We have supplemental materials (see Supplementary Note 1: HeALTH Design Documentation) detailing the overall design, circuit diagrams, bill of materials, and cost of materials for each component of the system. These materials allow others to replicate and build the system, as mentioned in lines 142-144. We have also added additional text in the manuscript (lines 312-317) discussing the general scalability and feasibility of the system in other labs.

“An overview of the design of each subsystem, along with a complete list of parts and cost to create the HeALTH system can be found in **Supplementary Note 1.**” (lines 142-144)

“Furthermore, the overall cost and relatively small physical footprint makes it feasible to create replicates of the system and expand experimental capacity without dramatically increasing space or large equipment requirements. In fact, many of the components used to create the system, such as the main body of the platform, are modified from commercially available parts or DIY kits. Coupled with the design documentation and parts list (detailed in **Supplementary Note 1**), the system could be recreated in other labs.” (lines 312-317)

4) Although mentioned, it's difficult to understand how this will be done. "We could also examine the effects of intermittent fasting, perform drug screens, or examine the effect of stressors (such as heat shock, oxidative stress, or osmotic stress)"

As mentioned in the results and discussion section of the manuscript, the design of the microfluidic device enables us to perform fluid exchange across the population of interest. By adding chemicals of interest to the liquid food source, the HeALTH system could easily expose populations to different drug compounds for drug screens, different concentrations of paraquat for examining oxidative stress, or concentrations of NaCl for osmotic stress assays. In fact, we already demonstrate the ability of our system to deliver drug compounds in a robust manner through using the small molecule of C22 within the liquid food source as a method of progeny prevention (see methods section). Similar to how we demonstrated the ability to perform dynamic environmental perturbations, we could also switch the food source experienced over time to explore the effects of intermittent fasting on lifespan and behavioral decline with age.

Lastly, as mentioned in the results section of the paper (Figure 1c), we are able to precisely control the thermal conditions experienced by the worms over time. Thus, we could easily subject the worms to acute heat stress (35°C), allowing us to examine how the individual behavior changes under thermally stressed conditions. Since these assays typically are completed within 24 hours, our system could easily be used to maintain precisely controlled thermal conditions over time, while simultaneously being able to monitor their behavioral response and survival over time.

REVIEWERS' COMMENTS:

Reviewer #1 (Remarks to the Author):

The resubmitted paper by Le & Zhan et al., "HeALTH: An Automated Platform for Long-term Longitudinal Studies of Whole Organisms under Precise Environmental Control," describes a novel microfluidic device for monitoring *C. elegans* behavior in a long-term and controlled manner. This is a comprehensive study of a useful device that will be of great interest to the aging field. The authors have adequately addressed reviewer comments and I enthusiastically recommend this work for publication.